# Genome Editing as a Treatment for the Most Prevalent Causative Genes of Autosomal Dominant Retinitis Pigmentosa

**DOI:** 10.3390/ijms20102542

**Published:** 2019-05-23

**Authors:** Michalitsa Diakatou, Gaël Manes, Beatrice Bocquet, Isabelle Meunier, Vasiliki Kalatzis

**Affiliations:** 1Inserm U1051, Institute for Neurosciences of Montpellier, 80 Avenue Augustin Fliche, 34091 Montpellier, France; michalitsa.diakatou@inserm.fr (M.D.); gael.manes@inserm.fr (G.M.); beatrice.bocquet@inserm.fr (B.B.); isabelannemeunier@yahoo.fr (I.M.); 2University of Montpellier, 34095 Montpellier, France; 3National Reference Centre for Inherited Sensory Diseases, CHU, 34295 Montpellier, France

**Keywords:** Inherited retinal dystrophies, autosomal dominant retinitis pigmentosa, photoreceptors, loss-of-function, gain-of-function, dominant-negative, CRISPR/Cas, gene supplementation, genome-editing, AAV vector

## Abstract

Inherited retinal dystrophies (IRDs) are a clinically and genetically heterogeneous group of diseases with more than 250 causative genes. The most common form is retinitis pigmentosa. IRDs lead to vision impairment for which there is no universal cure. Encouragingly, a first gene supplementation therapy has been approved for an autosomal recessive IRD. However, for autosomal dominant IRDs, gene supplementation therapy is not always pertinent because haploinsufficiency is not the only cause. Disease-causing mechanisms are often gain-of-function or dominant-negative, which usually require alternative therapeutic approaches. In such cases, genome-editing technology has raised hopes for treatment. Genome editing could be used to (i) invalidate both alleles, followed by supplementation of the wild type gene, (ii) specifically invalidate the mutant allele, with or without gene supplementation, or (iii) to correct the mutant allele. We review here the most prevalent genes causing autosomal dominant retinitis pigmentosa and the most appropriate genome-editing strategy that could be used to target their different causative mutations.

## 1. Inherited Retinal Dystrophies 

The retina is the tissue that lines the inner back of the eye. It is composed of two parts: the single-layered retinal pigment epithelium (RPE) on the posterior side, and the multi-layered neuroretina on the anterior side (Figure 1A). The neuroretina is composed of interconnecting layers of neuronal cells responsible for detecting the incoming light signal, transforming it into an electrical signal, and relaying this signal to the brain for image interpretation. The light-sensing cells of the neuroretina are the photoreceptors. These cells have a characteristic morphology comprising of a cell body and an inner segment, which is attached, via a connecting cilium, to an outer segment filled with lipid discs (Figure 1B). It is within these outer segment discs that the phototransduction process takes place. There are two types of photoreceptors that differ in the length of their outer segments and respond depending on the light intensity: rods are responsible for night and peripheral vision, and cones are responsible for day and fine vision. The photoreceptor outer segments are in close contact with the RPE, which is essential for photoreceptor survival and function by absorbing excess light, phagocytizing shed outer segments, secreting growth factors, removing water and ions, and providing nutrients and growth factors [1].

Due to the interdependence of the photoreceptors and the RPE, a dysfunction of either or both of these tissues results in visual defects. If this dysfunction is genetic in origin, it gives rise to a group of conditions referred to as Inherited Retinal Dystrophies (IRDs; Figure 2). This clinically and genetically heterogeneous disease group collectively affects 1 in 2000 people [2]. Although some stationary forms exist, most IRDs are characterized by progressive vision loss, which, in the absence of treatment, can lead to legal blindness [3]. The retinal phenotypes range from mild to severe and can be either isolated (non-syndromic forms) or associated with extra-ocular signs (syndromic forms). Non-syndromic IRDs are further categorized based on the region of the retina that is affected (Figure 2). When only the central retina (the macula) is affected, they are denoted as macular dystrophies [4]. When the whole retina is affected, the diseases are classified according to the photoreceptor type that degenerates first: if rods are first affected, they are referred to as rod-cone dystrophies [5]; if cones are first affected (or simultaneously with rods), they are referred to as cone-rod dystrophies [6]. In addition, cones can be exclusively affected, and the associated diseases are referred to as cone dystrophies. One particular group of disorders that falls outside of these categories is Leber congenital amaurosis (LCA), in which both the macula and peripheral retina are affected and degenerate rapidly from birth [7]. Lastly, when the choroid, the highly vascularized tissue situated behind the RPE that nourishes the retina, is also damaged, this group of IRDs is referred to as chorioretinopathies [4]. 

Distinct clinical features associated with certain IRDs can sometimes help to orient the genetic diagnosis. This is noteworthy, as each disease is monogenic and mutations in over 250 genes have been proven to be causative (see http://www.sph.uth.tmc.edu/Retnet). In addition, IRDs have autosomal recessive or dominant, X linked, and, more rarely, mitochondrial inheritance patterns [8]. To further complicate diagnosis, many genes give rise to more than one disorder. This is illustrated in Table 1, which lists all the genes that are implicated in the most common IRDs and highlights genes that are shared between different disorders. It can be noted that mutations in many genes can be transmitted both in a dominant and recessive manner. 

To date, there is still no cure for IRDs. Furthermore, the development of a single approach is not feasible due to the heterogeneity (age of onset, speed and severity of progression, cell type predominantly affected, type of cellular dysfunction) of each disorder. Although pharmacological molecules have been developed to slow disease progression, they do not constitute a real solution [9]. However, the monogenic nature of IRDs lends them to the development of gene-based therapies. Along this line, the retina has several key characteristics that render it as an ideal tissue for this type of therapy [10]. Firstly, it is highly accessible for imaging and surgery, secondly it is enclosed allowing the administration of small amounts of the therapeutic product, and lastly, it is sequestered from the systemic circulation by the blood-retinal-barrier [11], thus providing it with an immuno-privileged status. For all the aforementioned reasons, retinal gene therapy has been the focus of much interest over the last ten years. This field has progressed the fastest for autosomal recessive IRDs, which are amenable to straightforward gene supplementation therapy [12]. However, it is important to now address the challenges associated with autosomal dominant disorders, and the evolution in genome-editing strategies, provides hope for the treatment of these IRD forms.

Here, we focus on the autosomal dominant forms of the most common rod-cone dystrophy, retinitis pigmentosa (adRP). We present an overview of the more prevalent adRP causative genes and mutations, and discuss the most suitable genome-editing strategies. These examples serve as models that can be extrapolated beyond adRP to other autosomal dominant IRDs.

## 2. Mutation Type and Compatible Therapeutic Approaches

### 2.1. Types of Dominant Mutations

Dominant mutations can be broadly divided into three types based on their disease-causing mechanism [13]: 

*Loss-of-function:* mutations that render the product of a gene non-functional. Pathogenicity arises because one copy of the gene is not sufficient to assure a normal phenotype, a condition known as haploinsufficiency. Generally, this type of mutation causes autosomal recessive disorders [14]. Dominant mutations due to haploinsufficency are less frequent but they do occur, such as most mutations in the pre-mRNA processing factor 31 homolog gene, *PRPF31*, that causes adRP.

*Gain-of-function:* mutations that attribute a new function to a protein, which may be toxic to the cell. Typical examples are most of the mutations in the *RHO* gene. *RHO* encodes the light-sensitive rhodopsin protein, involved in phototransduction. Mutant rhodopsin molecules cannot function normally and, given the high expression levels of rhodopsin in photoreceptors, cell mechanisms such as trafficking or protein degradation can be overwhelmed by large quantities of the mutant forms [15].

*Dominant-negative:* mutations resulting in a mutant protein that interferes with the function of the wild type [16]. An example are mutations in the gene *RP1* [17,18]. RP1 is a photoreceptor-specific microtubule protein, important for the organization of membrane discs [8,10]. Shifting the ratio of the mutant p.Gln662* RP1 versus wild type protein in a knock-in mouse model delays photoreceptor regeneration [19].

It should be mentioned that not all mutations fit clearly into one of the above categories. For many mutations that cause autosomal dominant IRDs, the mode of action remains elusive or stems from a combination of the three classes. This is especially true for gain-of-function and dominant-negative mutations (specific examples are discussed below).

### 2.2. Gene Therapy Approaches

The type of autosomal dominant mutation associated with a particular disorder dictates the appropriate gene therapy approach. For loss-of-function mutations, higher protein levels are required. Therefore, supplementation of an extra copy of the wild type gene may be sufficient for phenotype restoration. In such cases, adeno-associated viral (AAV) vectors are most commonly used (Figure 3) due to their high transduction efficiency and excellent safety profile [12]. However, in cases where the cell is sensitive to the levels of the encoded protein, supplementation may not be suitable.

Gene supplementation alone has also been suggested for some cases of dominant-negative mutations [13,20,21]. The reasoning is that by increasing the ratio of wild type versus mutant protein, the mutant protein will be out-competed and some level of function will be restored. However, caution needs to be taken, as this approach may only alleviate symptoms but not necessarily represent a cure. A reason is that even though a mutation may behave primarily as dominant-negative, it may also have toxic side effects for the cell, which will not be lessened by the extra levels of wild type protein [13,22]. Therefore, generally, in order to treat dominant-negative as well as gain-of-function mutations, it is necessary to interfere with the mutant allele at the DNA or RNA level, prior to, or in lieu of, gene supplementation (for comprehensive review see [23,24]). Three basic strategies (Figure 3) have been explored in order to fulfil this purpose:

*a. Invalidation of both alleles and gene supplementation.* This is probably the most straightforward approach in terms of gene invalidation. The rationale is to design molecules that target both the mutant and wild type allele thus blocking protein production prior to supplying an exogenous gene copy. An example at the DNA level is transcriptional repression using engineered zinc finger proteins, which has been shown to successfully silence human rhodopsin [25,26]. At the RNA level, examples include the use of ribozymes [27] to mediate destruction of the target RNA, or RNA interference (si/shRNA) technologies [28] to down-regulate translation.

*b. Invalidation of the mutant allele.* It has been a longstanding goal of researchers to specifically target the mutant allele while sparing the wild type. The strategies can be the same as those for the invalidation of both alleles, but using molecules that specifically recognize the mutant allele. At the DNA level, genome editing by the relatively recent CRISPR/Cas system, which is detailed in a separate section below, has been used to introduce targeted double-strand breaks for specific allele ablation [29,30,31]. Examples for RNA destruction or interference include ribozymes [32] and anti-oligosense nucleotides [33], respectively.

*c. Correction of the mutant allele.* This more sophisticated approach couples genome-editing technologies with homologous recombination to specifically correct mutations at the DNA level. Examples include the use of zinc finger nucleases [34] and the CRISPR/Cas system [35] to induce a double-strand break and recombination of a donor DNA fragment (see the following section).

## 3. Genome Editing via the CRISPR/Cas System

As mentioned above, the latest and most promising tool for genome editing is the CRISPR/Cas system, short for Clustered Regularly Interspaced Short Palindromic Repeats (CRISPR) and CRISPR-associated gene (Cas) system [36,37,38]. The CRISPR/Cas system was originally developed by bacteria and archaea [39,40,41,42,43,44] to identify an intruding virus and induce double strand breaks in its DNA [45]. The bacteria then incorporate this library of viral DNA fragments with which they have been infected into their genome as a defense mechanism. This library corresponds to the CRISPR sequences. In this way, when the bacteria are re-infected by the same virus, double-strand breaks are induced in the viral DNA by the Cas nuclease. The specific recognition and cutting of the viral genome by the Cas nuclease are reliant on two sequences: the guide RNA (gRNA), a 20-nucleotide (nt) sequence homologous to the target sequence, and the following protospacer adjacent motif (PAM), which is composed of 3-nt [36]. 

Using the CRISPR/Cas system, we can theoretically target any DNA sequence. The only requirement is that the target sequence contains a PAM sequence. Each PAM sequence is recognized by the Cas nuclease of a particular bacterial species. The most commonly used is Cas9 from *Streptomyces pyogenes* (SpCas9), which recognizes the PAM NGG. The PAM sequence in the target DNA sequence will dictate which Cas nuclease can be used. Once the Cas nuclease is delivered into the cells, it will induce a double-strand break in the target sequence. The cell will then repair the breaks [46,47] using one of two main repair pathways: *non-homologous end-joining (NHEJ)* or *homology-directed repair (HDR)*.

During *NHEJ*, in order for the double-strand break to be repaired, the cell machinery joins the two ends of the break. However, this is an error-prone mechanism as insertions and deletions (indels) usually occur. These indels often induce a frameshift in the sequence, which would likely result in a premature termination codon. As a consequence, the corresponding gene is no longer functional [48]. This approach can be used to invalidate either the mutant allele specifically, if the gRNA only targets this allele, or both alleles (Figure 3). If Cas9 recognizes both alleles, then supplementation of the wild type gene is also required.

During *HDR*, the cell uses the sister chromatid of the homologous chromosome as a template to repair the break. Alternatively, if a wild type sequence, usually in the form of a single-stranded oligodeoxynucleotide (ssODN), is provided to the cell simultaneously with the Cas9, that sequence can be used as a template to promote HDR repair [49]. This would be the method of choice in order to precisely correct an allele (Figure 3). The limitation of HDR is that it occurs much less frequently than NHEJ in the late S and G2 phases of cellular division [50]. Therefore, as the photoreceptors and the RPE are post-mitotic cells, they lack the HDR mechanism. A way to circumvent this issue has been proposed by Suzuki and colleagues and is called HITI (Homology-Independent Targeted Integration). HITI uses the NHEJ mechanism, and not HDR, to integrate a DNA sequence. As a result, non-diving cells, such as photoreceptors, can be potentially edited using this method [51].

Thus, taken together, the highly specific and efficient CRISPR/Cas system shows much potential for the treatment of the genetically heterogeneous IRD group [52].

## 4. Genome Editing for Autosomal Dominant Retinitis Pigmentosa

Amongst IRDs, RP is in itself a large and genetically heterogeneous group of disorders. Therefore, we focus here on the most frequent causative genes and associated mutations for the autosomal dominant forms, and discuss potential targeted therapeutic strategies. 

RP has a prevalence of 1 in 4000 individuals and is the most characteristic clinical representation of rod-cone dystrophies [53]. It is characterized by a “tunnel vision”, resulting from progressive loss of peripheral vision. Central vision is usually maintained until the end stages of the disease [54]. RP can be transmitted with an autosomal dominant, recessive or X-linked mode of inheritance. Almost 90 genes have been associated with RP (https://sph.uth.edu/retnet/), 29 of which cause adRP (Table 1). Among these, 6 genes (*NR2E3*, *NRL*, *RHO*, *RP1*, *RPE65* and *BEST1*) cause both autosomal dominant and recessive forms. Furthermore, there is genetic overlap between RP and other retinopathies. In Figure 4, we can see that 6 genes are responsible for both autosomal dominant forms of RP and macular dystrophy or cone-rod dystrophy. We place a particular emphasis here on the 8 most prevalent causative genes of adRP, which are *RHO*, *PRPF31*, *RP1*, *PRPH2*, *IMPDH1*, *NR2E3*, *SNRPN200* and *CRX* [55,56].

### 4.1. RHO

The *RHO* gene encodes rhodopsin, a light-sensitive protein, which is involved in visual transduction. Rhodopsin is a G-coupled receptor, which comprises almost 50% of the total protein content of rod outer segments and 80% of that of discs [13,57]. *RHO* is the most common gene causing adRP, with the percentage of cases varying depending on the geographic area, up to 30% in the US [55,58] and between 16%–20% in Europe [56,59]. According to the Human Gene Mutation Database (http://www.hgmd.cf.ac.uk/ac/index.php), almost 220 *RHO* mutations have been identified to cause different forms of RP. The adRP-causing mutations disturb protein structure and diverse functions, including folding, trafficking, endocytosis, membrane homeostasis, and proteasome degradation, and are grouped into classes depending on the function that is affected [54]. 

The majority of *RHO* mutations that cause adRP are gain-of-function mutations [15]. A good example is the p.Pro23His (P23H) mutation, which is the most common *RHO* mutation associated with adRP. P23H belongs to the so-called class II RHO mutants, which do not fold properly [15,54]. Due to their improper folding, class II mutants are labeled with ubiquitin and are destined for degradation by the ubiquitin proteasome system (UPS) [60]. Because of the large protein load, the degradation machinery is overwhelmed, which results in a failure to clear other misfolded proteins and leads to cell toxicity. Furthermore, in some cases, wild type rhodopsin becomes trapped with the class II mutant protein in the endoplasmic reticulum [22]. As a result, the wild type protein is more highly cleared by the proteasome [61] and is not delivered to the outer disc membrane [61,62]. As such, we could say that class II mutants have a secondary dominant-negative effect.

Given the high prevalence of *RHO* mutations, and in particular P23H, it is not surprising that this gene has attracted the highest interest for gene therapy studies. As P23H partially exerts a dominant-negative effect, a simple strategy of supplementation with wild type rhodopsin has been studied. Mao et al. provided wild type *RHO* to P23H transgenic mouse using an AAV2/5 vector and showed reduced retinal degeneration up to 6-months post-treatment compared to non-treated controls [20]. However, one of the challenges of supplementation is that rods are highly sensitive to the levels of rhodopsin. Price and colleagues supplemented mice carrying the P23H allele with increasing numbers of wild type gene copies [21]. Although the authors confirmed that the dominant-negative effect of the mutant protein could be largely overcome by increasing the levels of the wild type protein, they reported that above three copies of the transgene, retinal deterioration persists; most likely due to a secondary effect of cell crowding. Thus, too low or too high levels of exogenous rhodopsin can lead to cell toxicity and retinal degeneration [21,63]. 

Consequently, more emphasis has been given to supplementing exogenous rhodopsin while silencing the endogenous gene. Several groups have studied this approach by RNA interference technology. One group used AAV 2/5 vectors to deliver siRNAs that targeted the endogenous P23H *Rho* allele while providing exogenous wild type *RHO* to a heterozygous *Rho*^+/−^ mouse model. The authors demonstrated improvement in ONL thickness up to 9 months post-injection [64]. Similarly, another group designed a study on a mouse model carrying another *Rho* mutation, the p.Pro347Ser (P347S) allele [65]. The authors used two AAV2/5 vectors: one to deliver the shRNA targeting the mutant allele and the second to deliver an extra copy of wild type *RHO*. Although an initial improvement was observed, the beneficial effect faded by 20 weeks post-injection. Lastly, Mitra and colleagues conducted a proof-of-concept study delivering shRNA and wild type *RHO* with nanoparticles to a knock-in *Rho^P23H/P23H^* mouse model and reported partial improvement of visual function [66]. Taken together, these studies showed moderate results. This may be explained by the fact that rhodopsin is an intensely active gene and it is challenging to reduce protein levels sufficiently by interfering at the mRNA level.

Recently, Tsai and colleagues used an alternative approach, which consisted of mutation-independent *Rho* ablation with the CRISPR/Cas9 system followed by gene supplementation. The authors tested this approach on two mouse models, one for the P23H mutation and one for the p.Asp190Asn mutation, using a dual AAV2/8 vector system to vehicle the CRISPR/Cas system and the exogenous *RHO* gene. Following ablation and replacement, ONL thickness was reported to be increased by up to 35% and accompanied by significantly improved ERG responses in both models, in contrast to gene supplementation alone [67].

Due to the partial success of these dual allele-independent gene therapy approaches, it is not surprising that many teams have taken advantage of the advent of CRISPR/Cas9 technology to explore allele-specific invalidation of *RHO*. Bakondi and colleagues were the first to target an allele-specific PAM sequence present only in the *Rho^S334^* mutant allele of an RP mouse model. Following subretinal administration and electroporation of the CRISPR/Cas components, the photoreceptor phenotype was rescued and visual acuity was increased by 53% [29]. Similarly, Latella et al. targeted a *RHO^P23H^* minigene expressed in a transgenic mouse model. They reduced expression of this mutant allele by subretinal electroporation of Cas9 and two gRNAs that targeted the 5′ and the 3′ regions of exon 1 [30]. More recently, Giannelli et al., used a variant of SpCas9 called VQR, which recognizes a different PAM sequence than wild type SpCas9, to selectively target the mutant allele in a *Rho^+/P23H^* mouse model. Following delivery of the CRISPR/Cas system by AAV2/9 vectors, the degeneration of photoreceptors was slowed and visual function was improved [68]. Lastly, Li et al., reported discrimination between the human wild type and mutant allele in the Rho-P23H mice using an improved version of SpCas9-VQR, called VRQR, and truncated gRNAs. They report that 45% of the mutant protein was edited at the DNA level and that photoreceptor degeneration was significantly delayed [69]. 

Although all these studies are at initial stages, they provide promise for the gene therapy of *RHO*-associated adRP. As CRISPR/Cas9 technology continues to advance it is certain that more studies will take place with interesting results. The particularly high prevalence of the P23H mutation, notably in the US population, renders these studies highly pertinent clinically.

### 4.2. PRPF31

The pre-mRNA splicing factor, PRPF31, is a ubiquitously expressed component of the spliceosome. It is the second most prevalent gene to cause adRP, accounting for up to 10% of all cases [55,56]. Although *PRPF31* mutations are lethal in the homozygous state [70], it is still largely unknown why they only cause retinal disease in the heterozygous state. One suggestion is that photoreceptors are more sensitive to PRPF31 levels due to their high demand on the spliceosome machinery [71]. Along this line, photoreceptors have been shown to follow a special splicing program that produces high levels of alternatively spliced transcripts in comparison to other retinal cells [72].

There have been 175 *PRPF31* variants identified to date. Most of these are either large deletions or result in a premature termination codon whereby the mutant transcript is cleared through nonsense-mediated decay (NMD) [73,74]. As large deletions are difficult to detect, the number of mutations may be underestimated. Mutant transcripts that are cleared by NMD are considered to be null alleles and the carriers are functional hemizygotes. As the onset of symptoms seems to correlate with protein levels, the mode of action of *PRP31* mutations appears to be haploinsufficiency [73].

A striking feature of *PRPF31* mutations is the associated incomplete penetrance within the same family. It is believed that disease manifestation is linked to the levels of the wild type protein. Three different factors that may affect these levels have been proposed: The first is *CNOT3*, which acts as a negative regulator of *PRPF31*. Lower expression of *CNOT3* leads to less inhibition of PRPF31 and no clinical signs of the disease. In turn, the amount of *CNOT3* expression seems to be linked with an intronic polymorphism [75]. The second factor is MSR1, a minisatellite repeat element, which is adjacent to the *PRPF31* core promoter [76]. When there is at least one higher expressing 4-copy repeat *MSR1* allele, then the carriers of the *PRPF31* mutation are asymptomatic. When the carriers only have a 3-copy repeat *MSR1* allele, then they can be either symptomatic or asymptomatic (ibid). Lastly, two *trans*-acting expression quantitative trait locus (eQTL), one situated on a different chromosome and the other located near *PRPF31* on the wild type allele, have also been proposed to modulate *PRPF31* expression [77]. 

Therefore, it seems that there are multiple mechanisms that regulate the expression of *PRPF31*, and that the amount of protein production is determinant for the development of the clinical signs. Consequently, there are theoretically two suitable approaches to treat the disease by gene therapy. The first is gene supplementation [78] but surprisingly, to our knowledge, there have been no such studies for *PRPF31*, even though the size of the open reading frame (1497 bp) is permissive for AAV-mediated gene transfer. However, a patent for such an approach has been filed (https://patents.google.com/patent/WO2016144892A1/en). A second approach would be to specifically correct the mutant allele by genome editing. Such an approach was used to target the pathogenic *PRPF31* mutation, c.1115–1125del11. This mutation was corrected in patient induced pluripotent stem cells (iPSC) using the CRISPR/Cas system and an ssODN as a template for HDR [79]. The corrected iPSC were then differentiated into photoreceptors and RPE, which showed improved cilia morphology and rescue of the phagocytic capacity, respectively, in comparison to untreated cells. Lastly, as there is not one prevalent *PRPF31* mutation, an interesting alternative could be to target a modulator of *PRPF31* (for example, to invalidate *CNOT3*), as this approach could potentially be applied to a larger subset of patients.

### 4.3. RP1

*RP1* is another highly prevalent gene for RP, whose prevalence differs geographically but can account for up to 10% of adRP cases [17]. RP1 protein is localized in the connecting cilium of photoreceptors. It is a photoreceptor-specific microtubule-associated protein (MAP) and it is necessary for the organization of the membrane discs [80]. To date, 190 RP1 mutations have been linked to RP, which are categorized into four classes. The majority of *RP1* mutations causing adRP are class II mutants [17,18]. These mutations are concentrated in a hotspot in exon 4 of *RP1* (nt position 500–1053) and result in premature stop codons due to either nonsense or frameshift mutations. The corresponding transcript escapes NMD but the resulting protein is missing a large part of the C terminal end.

Many of the dominant *RP1* mutations show incomplete penetrance, which depends on the patient’s genetic background. Haploinsufficiency has been excluded because carriers of null alleles are mostly asymptomatic, whereas homozygous individuals present with the disease [81]. The mechanism of action of *RP1* mutations is considered to be dominant-negative [17,18]. Mice homozygous for a dominant nonsense *RP1* mutation show outer segment disorganization and photoreceptor degeneration. However, this phenotype can be prevented by the expression of exogenous wild type RP1 [81]. It must be noted though that, like for rhodopsin, the levels of wild type protein need to be within a specific range, as too much RP1 will also result in degeneration [81]. 

Despite the high prevalence of this gene, to our knowledge there have been no other relevant gene therapy studies reported to date. In order to determine the most suitable approach, it is necessary to further elucidate the mechanisms of the disease. Nevertheless, currently, we can presume that all three approaches of gene therapy (correction of the mutant allele, invalidation of the mutant allele and/or wild type supplement) could be candidates for adRP associated with class II RP1 mutations.

### 4.4. PRPH2

The peripherin-2 gene, *PRPH2*, also known as retinal degeneration slow (RDS), participates in the formation of the photoreceptor outer segments. It produces a transmembrane glycoprotein, which is located at the rim of the discs and it is essential for their biogenesis and stabilization [82]. *PRPH2* is the most clinically heterogeneous gene among all non-syndromic IRD-causing genes. More than 180 mutations have been identified, causing autosomal dominant RP, macular degeneration and cone-rod dystrophy (Figure 4). Of these mutations, at least 50 are linked to adRP and account for 5%–10% of cases [83]. Some of this variability may result from modifier genes situated at another locus, one of which is considered to be *ROM1* [84]. It has been suggested that digenic RP can arise when the p.Leu185Pro mutation of *PRPH2* is inherited with heterozygous mutations in the gene *ROM1* [85,86].

As can be expected, not all *PRPH2* mutations have the same mode of action. Some of the mutations that cause adRP have a clear haploinsufficiency effect [82,87]. An example is the p.Cys214Ser (C214S) mutation, which is equivalent to a null allele, as the mutant protein is unstable and degraded. This is supported by the fact that the mouse *Rds*^+/−^ model, which also has one null *Prph2* allele, presents with a disease phenotype. This phenotype was rescued by expression of a wild type *Prph2* transgene in rods and cones [88]. Other mutations in *PRPH2* exhibit more complicated mechanisms of action. There are cases of gain-of-function mutations that behave in a dominant-negative fashion, such as the p.Pro216Leu (P216L) mutation. It has been shown in vivo that the amount of mutant P216L protein is only 8% of the wild type, and that it promotes degradation not only of the mutant but also of the wild type protein. The resulting low levels of protein lead to a phenotype similar to haploinsufficiency [89]. The retina is highly sensitive to the *PRPH2* levels, which need to be approximately 60%–80% of the normal levels [88,89].

It has been suggested that in the case of loss-of-function mutations, such as C214S, a gene augmentation approach can be appropriate [90,91]. However, the fact that the levels of *PRPH2* need to be finely tuned makes this approach challenging [82]. Furthermore, gene augmentation is probably not sufficient for treating gain-of-function mutations with dominant-negative effects [82]. Instead, the safest approach would probably be targeted gene therapy by either gene invalidation or gene correction.

### 4.5. IMPDH1 

The Inosine-5′-Monophosphate Dehydrogenase genes, *IMPDH*, encode an enzyme that catalyzes the conversion of inosine 5′-phosphate (IMP) to xanthosine 5′-phosphate (XMP), a step in the synthesis of guanine nucleotide. *IMPDH* is essential for cell growth, as inhibition of the enzyme leads to a stop in DNA synthesis [92]. There are two IMP dehydrogenases in humans encoded by the genes *IMPDH1* and *IMPDH2*. While all cells express both isozymes, retinal cells express only IMPDH1. Furthermore, there are two retina-specific isoforms of *IMPDH1*, each produced by alternative splicing [93,94].

Mutations in *IMPDH1* are found in 2.5%–3.5% of all adRP cases [55,56,95] and in some cases of autosomal dominant LCA type 11 [95]. In total, 24 mutations have been described, the vast majority of which are missense variants. The two most common *IMPDH1* mutations are p.Asp226Asn, comprising 2.2% of cases, and p.Asp311Asn, 1.5% of cases [55]. All *IMPDH1* mutations are found in the so-called CBS subdomain, which contains two cystathionine β-synthase repeats and shows the most structural differences between IMPDH1 and IMPDH2.

While all *IMPDH1* mutations are dominant, the mechanism of pathogenesis is unclear. There are different hypotheses on how these variants give rise RP, all of them related to decreased DNA binding. The first hypothesis is that IMPDH1 has a second role as a transcription repressor of histone genes and the E2F transcription factor by binding single-stranded DNA [96], and that adRP-causing mutations block this DNA binding [97]. The second is that IMPDH1 is associated with polyribosomes that translate rhodopsin, and that the causative mutations undermine this interaction [58]. The last hypothesis is that adRP mutations lead to structural changes in *IMPDH1* that render it constantly active, and lead to an imbalance in purine nucleotides [98].

As can be seen from above, *IMPDH1* is a very interesting gene, with multiple functions. However, to date, the way that it gives rise to RP is not clear. In the absence of further evidence, it would be safer to treat adRP-causing *IMPDH1* mutations either by correction of the mutant allele or by invalidation of both alleles and gene supplementation.

### 4.6. NR2E3

The Nuclear Receptor subfamily 2 Group E Member 3 gene, *NR2E3*, encodes a transcription factor mainly expressed in photoreceptors [99]. It is essential for the differentiation of rods [100]. NR2E3 belongs to the family of orphan nuclear receptors, as no ligand has yet been identified. It interacts with CRX, NR2D1 and NRL in rod precursors as well in mature cells [101]. This interaction promotes the transcription of rod-specific genes, such as rhodopsin. At the same time, NR2E3 suppresses the expression of cone-specific genes [102]. To date, 80 mutations have been reported for *NR2E3*. The majority of these mutations cause enhanced S cone syndrome, which is an autosomal recessive disorder [103]. Symptoms include vision loss with increased sensitivity to blue light. 

Interestingly, there is only one mutation in *NR2E3*, c.166G>A (p.Gly56Arg; G56R), that has conclusively been linked to adRP [104]. This one mutation is the second most common mutation for adRP (after the *RHO* P23H mutation) and accounts for 1%–2% of adRP in the US and 3.5% in Europe [104,105,106]. Therefore, one G56R-targeted strategy could potentially treat all *NR2E3*-related adRP patients. The glycine residue at position 56 of NR2E3 is located in its DNA-binding domain. When changed to an arginine residue, it has been proposed that the mutant protein cannot bind DNA or dimerize effectively with wild type NR2E3 [107]. However, the G56R NR2E3 protein can continue to interact with CRX. A hypothesis regarding its mechanism of action is that mutant NR2E3 recruits CRX but blocks its interaction with DNA. The G56R variant thus represents a gain-of-function mutation [108]. As a result, mutant NR2E3 cannot activate rhodopsin but it oversuppresses S and M cone opsins. This lack of rhodopsin would be detrimental for the cells, as mice with 50% reduction in rhodopsin show photoreceptor deterioration [109]. 

The small size of the *NR2E3* open reading frame (1230 bp), and the fact that there is only one mutation that causes adRP, which makes it an ideal target for gene therapy. As the mechanism of action seems to be gain-of-function, invalidation of the mutant allele, with or without gene supplementation, or specific correction of the mutant allele would be appropriate therapeutic approaches.

### 4.7. SNRNP200

Small Nuclear Ribonucleoprotein U5 Subunit 200 (*SNRPN200*) encodes BRR2, which is a component of the spliceosome. BRR2 is a RNA helicase that is a component of the U5 snRNP. BRR2, together with PRPF31, PRPF3, PRPF6 and PRPF8, belongs to the group of splicing factors that are part of the U4/U6-U5 tri-snRNP complex and which, when mutated, cause adRP [110]. 

*SNRNP200* is responsible for approximately 2.5% of all adRP cases [55,56]. In total, 35 *SNRNP200* mutations have been identified. Most of them are grouped within the first DExD box of the helicase, which is necessary for the unwinding of RNA [111,112]. *SNRNP200* mutations reduce helicase activity [113,114,115]. Furthermore, two *SNRNP200* adRP mutations (p.Ser1087Leu and p.Arg1090Leu) affect not only spliceosome activation but also proper recognition of splice sites [110,116,117]. 

The mechanism of pathogenicity of *SNRNP200* mutations is not clear but haploinsufficiency can probably be excluded as it has been reported that missing one allele is not pathogenic [118]. To our knowledge, no studies on gene therapy for this gene exist to date. Lacking more information, the safest therapeutic approach would be correction of the mutant allele.

### 4.8. CRX

One last causative gene that merits being mentioned is *CRX*, which encodes the cone-rod homeobox protein. CRX is a key player in the development and maintenance of photoreceptors. CRX regulates the transcription of several retinal genes and, in addition, is expressed in the pineal gland, where it regulates the circadian rhythm [119]. *CRX*, like *PRPH2*, is one of the genes that cause the most variable disease phenotypes. Over 100 *CRX* mutations have been identified, most of which cause autosomal dominant cone-rod dystrophy or LCA type 7 (Table 1 and Figure 4). Late onset *CRX*-related adRP cases have also been identified, which comprise 1.5% of all adRP [56]. In total, 5 *CRX* mutations that cause adRP have been identified to date: four missense mutations p.Ser152Tyr, p.Gly122Asp, p.Arg115Gln and p.Arg41Gln (the latter also causes cone-rod dystrophy), and one deletion of exons 3 and 4 [56].

Although there is variability in pathogenicity, it seems that *CRX* mutations are dominant-negative, gain-of-function or both. Haploinsufficiency does not seem to be the cause of disease as a carrier of a null allele was reported to have a normal clinical phenotype [120]. Strikingly, in this particular case, the patient’s daughter who was also heterozygous presented with LCA arguing for a more complicated mechanism. Furthermore, animal models heterozygous for null *Crx* mutations do not have the severe symptoms displayed by patients with dominant forms of the disease [121]. 

Consequently, in order to treat the adRP-related *CRX* missense mutations, either invalidation or correction of the mutant allele would be the strategy of choice. For the reported deletion, invalidation of both alleles, followed by gene supplementation, would be the only approach. Furthermore, if such an approach were successful, it could also be applied to the *CRX* point mutations indiscriminately.

## 5. Discussion/Insights 

The scope of this article was to provide an overview of how adRP could be treated in light of emerging technologies. For this purpose, we examined how genome-editing strategies have been or could be applied to the most common genes that cause adRP. We used adRP as an example because it is the most frequent disorder, but the same principles can be applied to other autosomal dominant IRDs, that have similar pathological mechanisms. 

When designing a genome-editing strategy, it is essential to take into account the pathological mechanism of the mutation of interest. At the same time, the mechanism of other mutations within the same gene is also informative. For example, if null alleles in the heterozygous state do not cause symptoms, this indicates that, regardless of the mutation, haploinsufficiency is not the cause of pathogenicity. Furthermore, it needs to be kept in mind that the mechanisms are not always black and white. Often, symptoms arise from a combination of mechanisms, such as gain-of-function and dominant-negative, which is the case for *RHO*. In these situations, it is better to opt for the safest approach, which would be invalidation of both alleles followed by gene supplementation or correction of the mutant allele. Wild type gene supplementation (without prior invalidation) on the other hand, is a safer option for cases of haploinsufficiency, such as for *PRPF31*.

Another crucial point is the threshold of protein levels in the cell. For some proteins there is a fine balance between insufficiency and toxicity (RHO) while for others, such as transcription factors, improper levels, whether they be too high or too low, could disrupt signaling pathways and cause secondary effects [122]. Therefore, following genome editing, it is important to assay the affected pathways for signs of imbalance. Lastly, it is desirable to develop therapies with as high an impact as possible. All mutations that cause IRDs are rare, some only affecting a few families worldwide, and it would not be economically viable to tailor a strategy for each one of them. Therefore, it is advantageous to design strategies that are mutation-independent and even disorder-independent. As a result, a maximum number of patients would benefit from the therapy.

CRISPR/Cas is currently the most promising tool for genome editing and it has dramatically changed the field of gene therapy. To date, for non-retinal diseases, CRISPR/Cas-mediated genome editing has already made its way to clinical trials, at least for ex vivo applications. For example, a phase 1/2 study using autologous CRISPR/Cas9-modified CD34+ human hematopoietic stem and progenitor cells for the treatment of beta thalassemia has already been approved (https://clinicaltrials.gov/ct2/show/NCT03655678). This is also the case for an ex vivo trial using autologous T-cells with an engineered T-cell receptor that recognize cancer cells (https://clinicaltrials.gov/ct2/show/NCT03399448). On the retinal landscape, the positive results on non-human primates for a CRISPR/Cas9 genome-editing strategy for *CEP290*, a gene that causes LCA type 10 [123], will likely soon result in the first genome-editing clinical trial for an IRD.

Aside from the more ‘traditional’ way of editing mutations with CRISPR/Cas9, there are new variants of this tool that can be envisaged for the treatment of IRDs. One example is CRISPR interference (CRISPRi) (Figure 3). CRISPRi uses the so-called dead Cas9 (dCas9) molecule, which has been mutated to have no nuclease activity [124]. The dCas9 molecule is coupled with a gRNA that targets the promoter of the gene of interest. As dCas9 cannot cleave DNA it remains bound to the promoter, preventing its transcription. To further promote suppression, dCas9 has been fused with transcriptional repressor domains [125]. The biggest advantage of this approach is its reversibility, as no permanent change is introduced in the DNA. CRISPRi could be an advantageous alternative to allele-specific invalidation if there exists a single nucleotide polymorphism in the promoter that distinguishes the mutant from the wild type allele. Otherwise, both promoters would need to be targeted and a wild copy of the gene provided. 

CRISPR activation (CRISPRa) is based on the same principle as CRISPRi [126,127]. For CRISPRa, dCas9, instead of being fused to a repressor, is fused to a transcription enhancer. In this way gene expression is enhanced. So far, applications have been focused on gene screening but we can envision this approach as an alternative to gene supplementation (Figure 3). A suitable case would be mutations in *PRPF31*. Enhanced transcription of both alleles would lead to higher levels of only the wild type protein, as mutant PRPF31 would still be cleared by NMD. Furthermore, CRISPRa would provide an advantage in the case of genes that are too large to be delivered by AAV vectors. 

Base editor is another exciting new technology as it allows the direct editing of specific bases. For the base-editor approach, an impaired Cas9 (i.e., a Cas9 that only induces a single-strand break, also called “nickase”) is fused to a cytidine deaminase, which directly edits C/G bases to T/A and A/T bases to G/C [128,129,130]. In this way, it is possible to specifically convert the mutated nucleotide to the wild type one. This would be an attractive alternative to the correction of a mutant allele, due to the low efficiency of HDR in the post-mitotic retinal cells (Figure 3).

Despite the advances in CRISPR/Cas9 technology, there are some limitations that remain and should be taken into consideration when designing a therapy. It is well known that Cas9 tolerates mismatches, which means that it induces double-strand breaks on sequences that differ by one or more base pairs from the gRNA sequence, leading to off-target effects [131]. Strategies that have been designed to circumvent this problem include the development of more specific Cas9 molecules and modifying the guide RNA design to increase specificity [132,133,134,135]. Another difficulty is that most Cas9 genes are too big to be delivered by a single AAV vector. However, smaller Cas9 molecules from different species have been discovered more recently that are small enough to be packaged within an AAV [136,137,138]. This opens up the possibility of directly administering the Cas9 gene into the retina for in vivo genome editing. 

To conclude, IRDs are currently incurable diseases but gene therapy holds great promise for developing a cure. This is evidenced by the fact that the first gene therapy drug for an IRD, Luxturna, an AAV2/2 vector carrying the *RPE65* gene for the treatment of RPE65-associated RP or LCA type 2 [139], has recently reached the market. It is certain that the field will grow exponentially in the future, especially with the aid of novel, CRISPR/Cas9-mediated, genome-editing strategies. Patients and researchers alike are awaiting the progress that is still to be made with excitement.

## Figures and Tables

**Figure 1 ijms-20-02542-f001:**
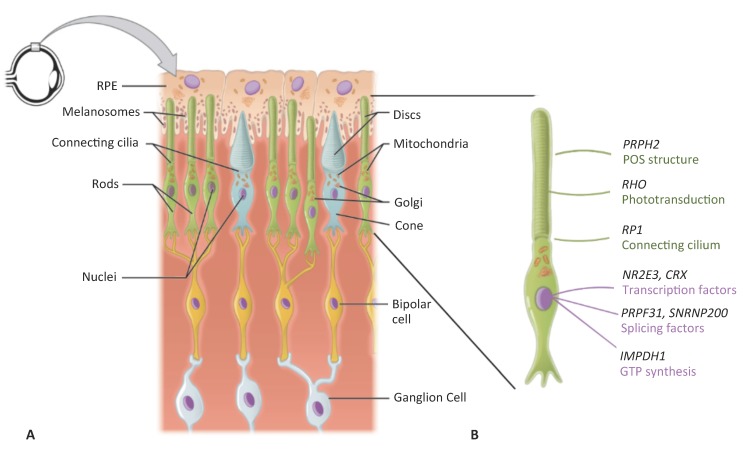
Schematic representation of the retina. (**A**) The mono-layered retinal pigment epithelium (RPE) is located on the posterior side of the retina. It contains apically located melanosomes that provide its pigmentation. The RPE is in close contact with the outer segments of the rod (in green) and cone (in blue) photoreceptors. Each outer segment, which contains the lipid discs important for phototransduction, is connected to the cell body of the photoreceptor by a connecting cilium. On the anterior side, the photoreceptors synapse with bipolar cells (in yellow), which in turn synapse with the retinal ganglion cells (in grey). (**B**) Higher magnification of a rod photoreceptor shown in A), depicting the characteristic rod structure and the site of action of the proteins encoded by the genes reviewed in this article. Modified from Wikimedia Commons (author OpenStax college). License to reproduce: https://creativecommons.org/licenses/by/3.0/legalcode.

**Figure 2 ijms-20-02542-f002:**
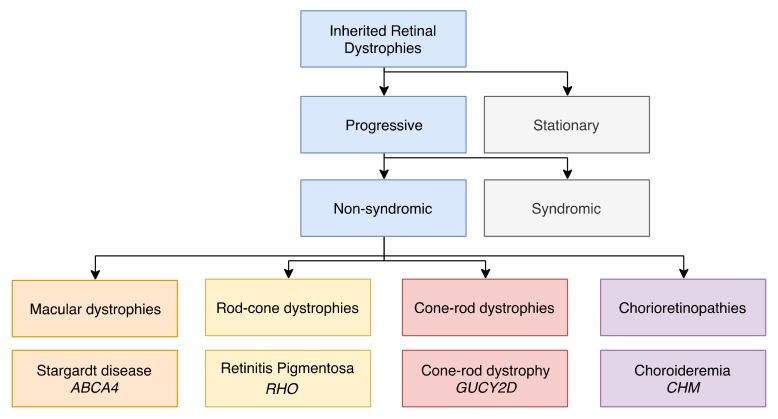
Subsets of inherited retinal dystrophies (IRDs). A chart showing the subdivision of progressive, non-syndromic IRDs into macular dystrophies, rod-cone dystrophies, cone-rod dystrophies and chorioretinopathies. Each class is illustrated by a main example of a retinal disorder and its causative gene.

**Figure 3 ijms-20-02542-f003:**
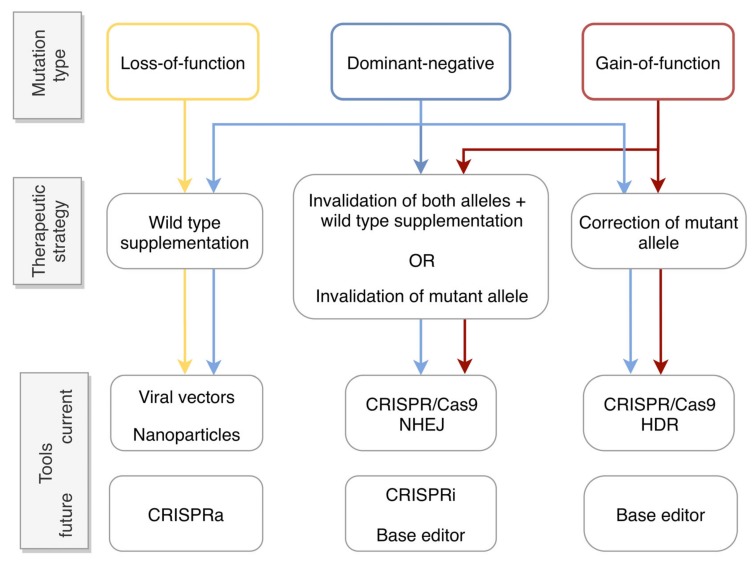
Flow chart showing the appropriate gene therapy strategy based on the type of mutation. Loss-of-function mutations (in yellow) could be treated by gene supplementation using, for example, viral vectors or non-viral nanoparticles to vehicle the transgene into cells. A future alternative would be to use CRISPR activation (CRISPRa) to enhance gene expression (see Section 5). Dominant-negative (in blue) and gain-of-function (in red) mutations can be treated by mutation-independent or -dependent gene invalidation using the CRISPR/Cas system and the error-prone non-homologous end-joining (NHEJ) repair pathway. In the case of mutation-independent invalidation, this would systemically need to be coupled to gene supplementation. The more recent CRISPR interference (CRISPRi) or base-editor technology, which can be used to suppress gene expression (see Section 5), also hold potential. Alternatively, these mutations can be treated by mutation-dependent correction using the CRISPR/Cas system and homology-directed repair (HDR). The recent base-editor technology also holds promise as a future gene correction strategy.

**Figure 4 ijms-20-02542-f004:**
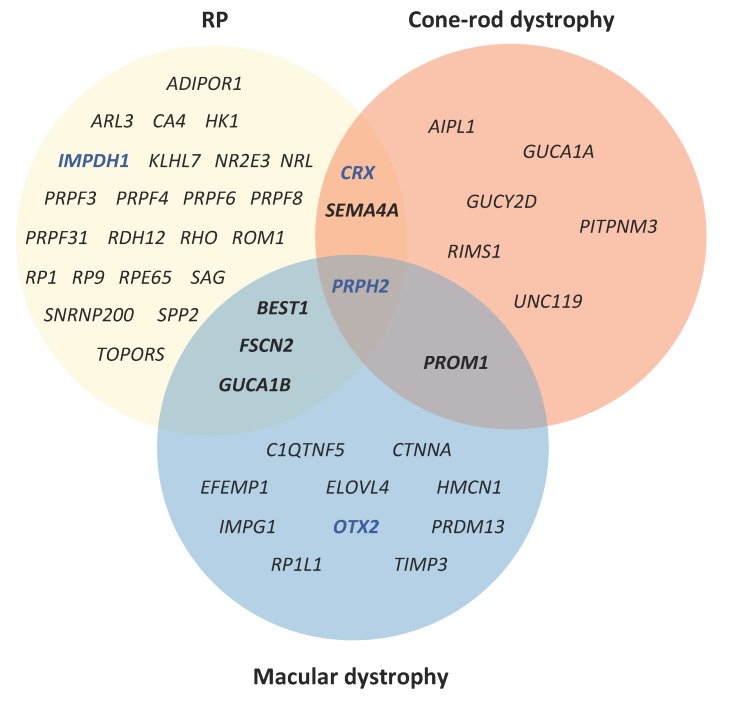
A Venn diagram of genes shared between different autosomal dominant IRD forms. Genes that cause RP are indicated in the yellow circle, genes that cause macular dystrophies are indicated in the blue circle, and genes that cause cone-rod dystrophies are indicated in the red circle. The genes *CRX* and *SEMA4A* cause both RP and cone-rod dystrophy, whereas the genes *BEST1*, *FSCN2* and *GUCA1B* cause both RP and cone-rod dystrophy. *PRPH2* is associated with all three forms. *PROM1* causes both autosomal dominant macula dystrophy and cone-rod dystrophy but not adRP. The genes indicated in blue (*IMPDH1*, *CRX* and *OTX2*) also cause autosomal dominant LCA.

**Table 1 ijms-20-02542-t001:** Genes associated with IRDs.

**Autosomal Dominant IRDs**										
Chorioretinal atrophy	*PRDM13*	*RGR*	*TEAD1*							
Cone(-rod) dystrophy	*AIPL1*	*CRX*	*GUCA1A*	*GUCY2D*	*PITPNM3*	*PROM1*	*PRPH2*	*RIMS1*	*SEMA4A*	*UNC119*
LCA	*CRX*	*IMPDH1*	*OTX2*							
Macular dystrophy	*BEST1*	*C1QTNF5*	*CTNNA1*	*EFEMP1*	*ELOVL4*	*FSCN2*	*GUCA1B*	*HMCN1*	*IMPG1*	*OTX2*
	*PRDM13*	*PROM1*	*PRPH2*	*RP1L1*	*TIMP3*					
RP	*ADIPOR1*	*ARL3*	*BEST1*	*CA4*	*CRX*	*FSCN2*	*GUCA1B*	*HK1*	*IMPDH1*	*KLHL7*
	*NR2E3*	*NRL*	*PRPF3*	*PRPF4*	*PRPF6*	*PRPF8*	*PRPF31*	*PRPH2*	*RDH12*	*RHO*
	*ROM1*	*RP1*	*RP9*	*RPE65*	*SAG*	*SEMA4A*	*SNRNP200*	*SPP2*	*TOPORS*	
**Autosomal Recessive IRDs**										
Cone(-rod) dystrophy	*ABCA4*	*ADAM9*	*ATF6*	*C21orf2*	*C8orf37*	*CACNA2D4*	*CDHR1*	*CERKL*	*CNGA3*	*CNGB3*
	*CNNM4*	*GNAT2*	*IFT81*	*KCNV2*	*PDE6C*	*PDE6H*	*POC1B*	*RAB28*	*RAX2*	*RDH5*
	*RPGRIP1*	*TTLL5*								
LCA	*AIPL1*	*CABP4*	*CCT2*	*CEP290*	*CLUAP1*	*CRB1*	*CRX*	*DTHD1*	*GDF6*	*GUCY2D*
	*IFT140*	*IQCB1*	*KCNJ13*	*LCA5*	*LRAT*	*NMNAT1*	*PRPH2*	*RD3*	*RDH12*	*RPE65*
	*RPGRIP1*	*SPATA7*	*TULP1*							
Macular dystrophy	*ABCA4*	*CFH*	*DRAM2*	*IMPG1*	*MFSD8*					
RP	*ABCA4*	*AGBL5*	*ARHGEF18*	*ARL6*	*ARL2BP*	*BBS1*	*BBS2*	*BEST1*	*C2orf71*	*C8orf37*
	*CERKL*	*CLRN1*	*CNGA1*	*CNGB1*	*CRB1*	*CYP4V2*	*DHDDS*	*DHX38*	*EMC1*	*EYS*
	*FAM161A*	*GPR125*	*HGSNAT*	*IDH3B*	*IFT140*	*IFT172*	*IMPG2*	*KIAA1549*	*KIZ*	*LRAT*
	*MAK*	*MERTK*	*MVK*	*NEK2*	*NEUROD1*	*NR2E3*	*NRL*	*PDE6A*	*PDE6B*	*PDE6G*
	*POMGNT1*	*PRCD*	*PROM1*	*RBP3*	*REEP6*	*RGR*	*RHO*	*RLBP1*	*RP1*	*RP1L1*
	*RPE65*	*SAG*	*SAMD11*	*SLC7A14*	*SPATA7*	*TRNT1*	*TTC8*	*TULP1*	*USH2A*	*ZNF408*
	*ZNF513*									

Genes mapped and identified for autosomal dominant and recessive forms of IRDs (data derived from RetNet, July 2018). The colored boxes indicate the genes (one color per gene) that are responsible for more than one type of disease. X-linked, stationary and syndromic IRDs, as well as developmental and mitochondrial disorders, have not been included.

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
