# Peer review of "Genome Editing as a Treatment for the Most Prevalent Causative Genes of Autosomal Dominant Retinitis Pigmentosa"

_ijms, 2019, doi:10.3390/ijms20102542_

Round 1

Reviewer 1 Report

The authors review the most prevalent dominant IRD genes and the most appropriate genome-editing strategy well. These informations can be used to target the different causative mutations in the future.

Author Response

We thank the reviewer for his positive review.

Reviewer 2 Report

The dysfunction of the photoreceptors and/or the retinal pigment epithelium (RPE) of the retina leads to Inherited Retinal Dystrophies (IRDs), clinically and genetically heterogeneous disease group which affects 1 in 2000 people. This heterogeneity poses major problem to the development of efficient therapies and to date, there is still no cure for IRDs. However, the monogenic nature of IRDs combined with the relatively easy and safe (due to immuno-privileged status of the retina) diagnostic and treatment accessibility of the retinal tissue make IRDs excellent target for gene-based therapies. Indeed autosomal recessive IRDs were already successfully treated by straightforward gene supplementation therapy. However the challenge of autosomal dominant disorders remains open as here more complex mechanisms are involved (apart from haploinsufficiency). In this latter case the genome-editing technology bears therapeutic promise due to its diverse capabilities which allow to invalidate both alleles (followed by wild type gene supplementation), specifically invalidate the mutant allele (with or without gene supplementation), or to correct the mutant allele. The review summarizes the state of the art knowledge of the most prevalent autosomal dominant IRD genes and the most relevant genome-editing strategies that can target the related causative mutations. 139 publications are referenced, ranging from classical to very recent (2017-2019), and produced by different labs, which guarantees the objective presentation of the current status of the field. The types of dominant mutations are thoroughly summarized in point 2.1 of the manuscript and the possible genome-editing treatment strategies are reviewed in detail in points 2.2.-4. Section 5 Discussion/Insights properly emphasizes that designing a genome-editing strategy, requires to take into account the pathological mechanism of the mutation of interest. The authors objectively identify that despite the advantages of CRISPR/Cas9 technology compared to the rest of the options, there are number of limitations that need to be addressed in order to make this approach clinically applicable, namely the potential for adverse effects (due to impacts on RNA structure) and due to the need to identify smaller Cas9 molecules that allow for AAV delivery.

The ongoing intensive work in the field clearly indicates that these goals are actively targeted by numerous groups and might be possible to be addressed in foreseeable future which makes the review a timely summary and roadmap for the future development of the field.

I have just a minor technical comment. Please increase the font size of the text in the boxes at Fig. 3 to improve its readability.

Author Response

We thank the reviewer for his/her positive evaluation. The font size of figure 3 has been modified according to his/her suggestion.

We took this opportunity to also increase the font size of Figure 1, and to make Figures 2 and 4 smaller. Moreover, we corrected some small typographical errors, and re-worded some sentences in the text to increase clarity. These changes have been highlighted in yellow. Lastly, we altered the Table format so that the lines are uniform in size. As a result, the whole table now fits on one A4 page, which facilitates reading.